# *PFN2a* Suppresses C2C12 Myogenic Development by Inhibiting Proliferation and Promoting Apoptosis via the p53 Pathway

**DOI:** 10.3390/cells8090959

**Published:** 2019-08-23

**Authors:** Huaqin Li, Lianjie Hou, Yu Zhang, Fangyi Jiang, Yifan Zhu, Qing X. Li, Ching Yuan Hu, Chong Wang

**Affiliations:** 1National Engineering Research Center for Breeding Swine Industry, Guangdong Provincial Key Lab of Agro-Animal Genomics and Molecular Breeding, College of Animal Science, South China Agricultural University, Guangzhou 510642, China; 2Department of Molecular Biosciences and Bioengineering, University of Hawaii at Manoa, 1955 East-West Road, Honolulu, HI 96822, USA; 3Department of Human Nutrition, Food and Animal Sciences, University of Hawaii at Manoa, 1955 East-West Road, Honolulu, HI 96822, USA

**Keywords:** proliferation, myogenic development, skeletal muscle, apoptosis, HDAC1

## Abstract

Skeletal muscle plays a crucial role in physical activity and in regulating body energy and protein balance. Myoblast proliferation, differentiation, and apoptosis are indispensable processes for myoblast myogenesis. Profilin 2a (PFN2a) is a ubiquitous actin monomer-binding protein and promotes lung cancer growth and metastasis through suppressing the nuclear localization of histone deacetylase 1 (HDAC1). However, how *PFN2a* regulates myoblast myogenic development is still not clear. We constructed a C2C12 mouse myoblast cell line overexpressing *PFN2a*. The CRISPR/Cas9 system was used to study the function of *PFN2a* in C2C12 myogenic development. We find that *PFN2a* suppresses proliferation and promotes apoptosis and consequentially downregulates C2C12 myogenic development. The suppression of *PFN2a* also decreases the amount of HDAC1 in the nucleus and increases the protein level of p53 during C2C12 myogenic development. Therefore, we propose that *PFN2a* suppresses C2C12 myogenic development via the p53 pathway. Si-*p53* (siRNA-*p53*) reverses the *PFN2a* inhibitory effect on C2C12 proliferation and the *PFN2a* promotion effect on C2C12 apoptosis, and then attenuates the suppression of *PFN2a* on myogenic differentiation. Our results expand understanding of *PFN2a* regulatory mechanisms in myogenic development and suggest potential therapeutic targets for muscle atrophy-related diseases.

## 1. Introduction

Approximately 45% of the human body mass is skeletal muscle [1]. Disease, injury, and aging can trigger skeletal muscle loss. Loss of muscle mass leads to decreased ability to resist disease and infection, and slowed wound healing, eventually leading to an inferior quality of life [2]. Skeletal muscle loss increases the cost of health care [3]. Muscle mass loss also decreases the protein and energy availability throughout the body [2]. Since skeletal muscle plays a crucial role in physical activity and regulating body energy and protein balance, it is essential to understand the mechanism of skeletal muscle myogenic development for preventing adverse effects on health and economic consequences.

Skeletal muscle regeneration and maintenance [4,5] rely on the quiescent skeletal muscle satellite cells, which exist between the sarcolemma of the muscle fiber and the basal lamina [6]. Depletion of skeletal muscle satellite cells in mice impairs muscle regenerative capacity [7]. Stimulated by injury or growth signals [8], skeletal muscle satellite cells activate and proliferate to generate myoblasts. These myoblasts undergo differentiation to either repair damaged muscle fiber or fusion to form new myofibers [9,10]. Recent studies have also found that apoptosis regulates the number of muscle cells and mediates myogenesis [11,12]. Muscle cell apoptosis results in skeletal muscle atrophy and sarcopenia in humans and rodents [13]. The differentiation of myoblasts is a complex multi-step process, while apoptosis is one of the routes accompanying differentiation [12]. Various apoptotic factors are involved in activating myogenic differentiation [14,15]. Elucidating the mechanism of muscle cell apoptosis is critical for understanding skeletal muscle development.

*Profilins* (PFNs) are actin-binding proteins and regulate the cell structure by regulating signal-dependent actin polymerization [16]. The *profilin* (*PFN*) gene family contains two major isoforms, *PFN1* and *PFN2* [17]. *PFN2* was alternatively spliced into *PFN2a* and *2b* in mice. *PFN2a* is the major splice form of *PFN2* [18,19] and is conserved among different vertebrates, such as humans, mice, chickens, and cattle [20]. *PFN2a* expresses in the mouse brain, testis, kidney, liver, and skeletal muscle [21]. Research on the function of *PFN2a* has focused on cell migration [22] and the mammalian nervous system, such as synaptic vesicle exocytosis and neuronal excitability [23]. However, little research has been done on muscles. Loss of *PFN2a* reduces the size of focal contacts and the number of migrating cells in chicken fibroblasts [20]. *PFN2a* overexpression in *Drosophila* cardiomyocyte induces cardiomyopathy [24]. *PFN2a* overexpression in *Drosophila* indirect flight muscles (IFM) reduces climbing ability, diminishes flight ability, and elongates thin filaments [24]. The *profilin* expression is decreased during the progression of C2C12 myogenic differentiation [25]. Those studies indicate that *profilins* play a critical role in myogenic development. The molecular mechanism by which *PFN2a* regulates muscle development, however, remains unclear. PFN2a regulates lung cancer growth through suppressing the nuclear localization of histone deacetylase 1 (HDAC1) [26]. Another study found that HDAC1 affects the activity of p53 by changing the p53 acetylation state and finally inducing p53 degradation, with alterations of the p53 target gene [27], and participates in cell growth and apoptosis. To our knowledge there is no published paper on the regulatory relationship between PFN2a and p53. The objective of this study was to elucidate the functions and regulatory mechanism of *PFN2a* in C2C12 myogenic development, and further enrich the regulation network of muscle development and regulation.

In this study, we constructed a *PFN2a*-overexpressing C2C12 cell line using the CRISPR/Cas9 system. Using immunofluorescence, laser scanning confocal, and gene interference, we found that *PFN2a* suppresses C2C12 myogenic development by inhibiting proliferation and promoting apoptosis via the p53 pathway. This study not only furthers our understanding of *PFN2a* function and regulatory mechanisms in myogenic differentiation but also provides experiment data for the future development of new strategies for treating muscle mass loss.

## 2. Materials and Methods

### 2.1. C2C12 Cell Culture, Transfection, and Differentiation

The C2C12 cell line (ATCC^®^ CRL-1772™) used in this study was purchased from American Type Culture Collection (ATCC, VA, USA). C2C12 cells were cultured in DMEM/HIGH GLUCOSE (Catalog No. SH30243.01, Hyclone, GE Healthcare Bio-Sciences, Pittsburgh, PA, USA) with 10% Fetal Bovine Serum (FBS) (Catalog No. FBS10099-141, Gibco, Grand Island, NY, USA). C2C12 cells (F2) were seeded in 6-well plates (2 × 10^4^/cm^2^). After 24 h, MCP-*ROSA26*-CG01 vector was transfected into C2C12 with DC-DON-SH02 (negative control), DC-DON-SH02-*PFN2a* (*PFN2a* donor), and DC-RFP-SH02 (positive control), respectively. The medium was replaced with new growth medium 6 h later, and cells were maintained in the growth medium for an additional 48 h before puromycin added. When we studied the function of *PFN2a* in C2C12 differentiation, WT (wild type C2C12 cells) and *PFN2a*-overexpressing C2C12 cells were seeded at same high density in 6-well plates (8 × 10^4^/cm^2^). When these cells were attached, we cultured in the differentiation medium (2% house serum) (Catalog No. SH30074.03, Hyclone, GE Healthcare Bio-Sciences, Pittsburgh, PA, USA) for seven days. As shown in Appendix A, si-*p53* (siRNA-*p53*) (5′-GAATGAGGCCTTAGAGTTA-3′) and si-NC (siRNA-negative control) were transfected into *PFN2a*-overexpressing C2C12 cells for 24 h to determine si-*p53* interference efficiency using Western blot and qPCR analyses. For RNA oligonucleotides, a concentration of 100 nM was used.

### 2.2. Construction of a PFN2a-Overexpressing Cell Line by CRISPR/Cas9

We used C2C12 cells (F2) to construct a *PFN2a*-overexpressing cell line. We constructed a *PFN2a*-overexpressing cell line by inserting a *PFN2a* transgene expression cassette into the genome *ROSA26* locus using the CRISPR/Cas9 system. The GeneHero™ mouse *ROSA26* safe harbor gene knock-in kit was purchased from GeneCopoeia Inc (Catalog No. SH-ROS-K200, GeneCopoeia Inc., Rockville, MD, USA). An MCP-*ROSA26*-CG01 vector was transfected into C2C12 with DC-DON-SH02, *PFN2a* donor, and DC-RFP-SH02, respectively. After transfection for 48 h, puromycin (2 μg/mL) was used to screen *PFN2a*-overexpressing monoclonal cells. After puromycin screening for 7 days, *PFN2a*-overexpressing monoclonal cells were obtained using cloning loops. Primers used to construct vectors and the quantitative analysis are shown in Table 1.

### 2.3. Identification of a PFN2a-Overexpressing Cell Line

We used the following three methods to identify the *PFN2a*-overexpressing cell line. We used *PFN2a*-overexpressing cells from the F3 generation for the correlation study. First, we used F2R2 primer to amplification of the ORF of *PFN2a*. Accurate integration of the exogenous *PFN2a* donor into the C2C12 genome *ROSA26* locus was performed. Primer sets of 5’HR (homology arms, HR) and 3’HR are composed of one primer within *ROSA26* genome (outside of the homology arms) and one primer within the donor transgene, to confirm on-target insertions (Figure 1B,C). Secondly, we used F3R3 primer to analyze the genotype of *PFN2a*-overexpressing monoclonal cells by PCR (Figure 1B,D). In addition, we used F4R4 primer to analyze the DNA copy number of *PFN2a* in *PFN2a*-overexpressing monoclonal cells by absolute quantitative PCR analysis (Figure 1E). Finally, Western blot was used to confirm the PFN2a protein level. Primers used to identify the *PFN2a*-overexpressing cell line are shown in Table 2.

### 2.4. RNA Extraction, Complementary DNA (cDNA) Synthesis, and Quantitative Real-Time PCR (qPCR)

Methods used for the RNA extraction and PCR analysis have been described previously [1]. For mRNA expression analysis, cDNA synthesis for mRNA was performed using HiScript® II Q RT SuperMix for qPCR (+gDNA wiper) (Catalog No. R223-01, Vazyme, Nanjing, China). qPCR was performed on a Bio-Rad CFX96 Real-Time Detection System (Bio-Rad, Hercules, CA, USA) using HieffTM qPCR SYBR^®^ Green Master Mix (NO Rox) (Catalog No. 11201ES08, Yeasen, Shanghai, China). The relative mRNAs level was normalized with *β-actin* level and indicated by 2^−ΔΔCt^. Primers used for qPCR analysis are shown in Table 3.

### 2.5. 5-Ethynyl-2′-deoxyuridine (Edu) Assays

We used the F4 generation of WT and *PFN2a*-overexpressing cells for EDU assays. WT and *PFN2a*-overexpressing cells were seeded in the 48-well plates at 2 × 10^4^/cm^2^, respectively. After 24 h of incubation in growth medium (GM), these C2C12 cells were used for Edu labeling by Cell-Light EdU Apollo567 In Vitro Kit (C10310-1, RiboBio, Guangzhou, China) according to the manufacturer’s instructions. The EdU-stained cells were visualized by using a Nikon TE2000-U inverted microscope (Nikon Instruments, Tokyo, Japan). Myoblast proliferation (ratio of EdU^+^ to all myoblasts) was counted using Nikon Instruments.

### 2.6. Flow Cytometry Analysis of the Cell Cycle

We used the F4 generation of WT and *PFN2a*-overexpressing cells for these studies. WT and *PFN2a*-overexpressing cells were seeded in the 6-well plates at 2 × 10^4^/cm^2^, respectively. After 24 h of incubation in growth medium (GM), these cells fixed in 70% ethanol overnight at −20 °C. Then these cells were performed propidium iodide (PI) staining by the Cell Cycle Detection Kit (Catalog No. KGA512, KeyGEN BioTECH, Guangzhou, China) according to the manufacturer’s instructions. These cells were analyzed by using Guava^®^ easyCyte™ Flow Cytometers (Merck KgaA, Darmstadt, Germany) and FlowJo 7.6 software (Tree Star Inc., Ashland, OR, USA).

### 2.7. Flow Cytometry Analysis of the Cell Apoptosis during C2C12 Myogenic Development

We used the F4 generation of WT and *PFN2a*-overexpressing cells for these studies. WT and *PFN2a*-overexpressing cells were seeded in the 6-well plates at a density of 2 × 10^4^/cm^2^ and cultured in growth medium (GM), respectively. After 24 h, we collect cells for proliferative phase apoptosis detection. WT and *PFN2a*-overexpressing cells were seeded in the 6-well plates at a density of 8 × 10^4^/cm^2^ and cultured in growth medium (GM), respectively. When the cells were attached, the medium was replaced with differentiation medium (2% house serum) for induction differentiation. After differentiation for 3 days, we collected cells for apoptosis assay using Annexin V-APC/7AAD apoptosis detection kit (Catalog No. AP105-100-KIT, Multisciences (Lianke) Biotech, Hangzhou, China), according to the manufacturer’s instructions. Apoptotic cell analysis performed BD AccuriC6 flow cytometer (BD Biosciences, San Jose, CA, USA) and FlowJo 7.6 software (Tree Star Inc.).

### 2.8. Western Blot Analysis

The method used for the Western blot analysis has been described previously [28]. Band intensities were quantified by Image J software. The antibodies and their dilutions used in this study are listed in Table 4 and Table 5.

### 2.9. Isolation of Nuclear and Cytoplasmic Extracts

We used the F5 generation of WT and *PFN2a*-overexpressing cells for these studies on proliferation day 1 and differentiation day 3. We separately collected WT and *PFN2a*-overexpressing cells on proliferation day 1 and differentiation day 3. The nuclear and cytoplasmic extractions were prepared using a NE-PER^TM^ Nuclear Cytoplasmic Extraction Reagent Kit (Catalog No. 78833, Pierce, Rockford, IL, USA) according to the manufacturer’s instruction. We measured the protein level of HDAC1 in the nuclear and cytoplasmic extract by Western blot.

### 2.10. Immunofluorescent Staining and Confocal Microscopy

We used the F4 generation of WT and *PFN2a*-overexpressing cells for these studies on differentiation day 7. WT and *PFN2a*-overexpressing cells were seeded the same high density (8 × 10^4^/cm^2^) in the 48-well plates and cultured in the differentiation medium for MyHC immunofluorescent assay. The method used for MyHC immunofluorescent assay analysis in 48-well plates has been described previously [1]. For the analysis of sarcomere structure, WT and *PFN2a*- overexpressing cells were seeded in the 6-well plates (8 × 10^4^/cm^2^) after adding Fisherbrand™ Microscope Cover Glasses (Catalog No.12-542A, ThermoFisher, Waltham, MA, USA), respectively. When the cells were attached, we changed the growth medium to a differentiation medium for induction differentiation. On differentiation day 7, the cells were fixed in 4% paraformaldehyde for 30 min and then washed three times with PBS for 5 min. Subsequently, the cells were permeabilized by adding 0.5% Triton X-100 (Catalog No. T8787, Sigma-Aldrich, Louis, MO, USA) for 30 min and blocked with 5% bovine serum albumin solution (BSA) (Catalog No. AR0004, BOSTER, Wuhan, China) for 2 h at 4 °C. Then, the cells were incubated with MyHC antibody and α-actinin antibody overnight at 4 °C. The Goat Anti-rabbit IgG/Alexa Fluor 647 and Goat Anti-Mouse IgG (H+L) Cy3 were added, and the cells were incubated at room temperature for 2 h. The cell nucleus was stained with DAPI (Catalog No. C1005, Beyotime, Jiangsu, China) for 30 min. Images were obtained with Leica confocal laser scanning microscope SP8 (Leica, Wetzlar, Germany).

### 2.11. Primers

All primers used in this study were designed by Premier Primer 5.0 software (Premier Bio-soft International, Palo Alto, CA, USA) and synthesized by TSINGK Biological Technology (TSINGK Biological Technology, Guangzhou, China). Information on all the primers used in this study is listed in Table 1, Table 2 and Table 3.

### 2.12. Statistical Analysis

All data are expressed as the mean ± standard error of the mean (S.E.M.) of three independent experiments. Our data is a normal distribution, and the homogeneity of data between each treatment group is equal under the SPSS analysis. In Figure 2, Figure 3, Figure 4 and Figure 5, and Appendix A, the unpaired Student’s *t*-test was used for *p*-value calculations, where * is *p* < 0.05; ** is *p* < 0.01; and *** is *p* < 0.001. In Figure 6 and Appendix A, one-way ANOVA (SPSS v18.0, IBM Knowledge Center, Chicago, IL, USA) was used for *p*-value calculations. We considered *p* < 0.05 to be statistically significant. In Figure 1F and Figure 5G,H, band intensities were quantified by Image J software (National Institutes of Health, Bethesda, MD, USA) and normalized to β-actin, tubulin or lamin B1. Data were expressed as change in fold relative to the control. In Figure 1, Figure 2, Figure 3, Figure 4, Figure 5 and Figure 6, WT is the abbreviation of wild type C2C12 cells. *PFN2a* cell is the abbreviation of *PFN2a*-overexpressing C2C12 cell.

## 3. Results

### 3.1. PFN2a Knock-in at ROSA26 Locus of C2C12 Cells was Made with CRISPR/Cas9

*PFN2a* has low mRNA level and protein level (Appendix A) during the progression of C2C12 myogenic differentiation. To elucidate the *PFN2a* function in C2C12 myogenic development, we constructed stable *PFN2a*-overexpressing C2C12 cells (*PFN2a* cell) using the CRISPR/Cas9 system. Immunofluorescent results showed successful insertion of *PFN2a* into C2C12 cells at *ROSA26* locus using CRISPR/Cas9, which offered candidate *PFN2a*-overexpressing monoclonal cells (Figure 1A). Next, we designed primers for screening monoclonal cells (Figure 1B). Results of 5’ homologous recombination (HR) and 3’HR primers PCR amplification showed the expected sizes of the expected DNA fragments, indicating correct integration of the *PFN2a* donor at the *ROSA26* locus (Figure 1C). Results of F2R2 PCR amplification showed existence of an open reading frame (ORF) of *PFN2a* in the *ROSA26* locus (Figure 1C). Genotype was determined via F3R3 PCR using DNA prepared from wild-type (WT) and monoclonal C2C12 cells. Results of PCR amplification showed F3R3 primers amplified a 354 bp wild-type fragment in the WT cells. F3R3 primers also amplified a 354-bp wild-type fragment, and a 4163-bp-containing *PFN2a* donor in monoclonal cells (Figure 1D). DNA copy number alteration of *PFN2a* increased by 0.5 times in monoclonal C2C12 cells compared with the WT cells (Figure 1E). Results of PCR amplification indicated that monoclonal C2C12 cells represent a heterozygous C2C12 *PFN2a*-overexpressing cell line. Western blot results showed that this monoclonal cell successfully overexpressed PFN2a (Figure 1F). A *PFN2a*-overexpressing C2C12 cell line was successfully constructed with the CRISPR/Cas9 system. This cell line was used to study the function of *PFN2a* in C2C12 myogenic development.

### 3.2. PFN2a Overexpression Suppressed C2C12 Proliferation

To investigate the role of *PFN2a* in C2C12 proliferation, we performed EDU labeling and cell cycle analysis. *PFN2a* reduced (*p* < 0.01) the percentage of Edu-positive cells (Figure 2A). Cells in S-phase mainly conduct DNA replication and prepare for cell division. Cell cycle analysis showed averages of 53%, 24%, and 21% cells for G1, S, and G2, respectively, in WT. However, the averages of 57%, 14%, and 27% cells were found in G1, S, and G2, respectively, in *PFN2a*-overexpressing cells. These data showed that *PFN2a* increased the percentage of cells in G1-phase (*p* < 0.01) and reduced (*p* < 0.05) the percentage of cells in S-phase (Figure 2B). Cyclins and p21 control the cell cycle progression. qPCR results showed that *PFN2a* significantly downregulated the mRNA level of related genes that regulate the cell cycle, such as cyclin B1 (*CCNB1*) (*p* < 0.05), cyclin D1 (*CCND1*) (*p* < 0.01), and proliferated cell nuclear antigen (*PCNA*) (*p* < 0.05), and significantly increased the (*p* < 0.01) *p21* mRNA level (Figure 2C). Western blot results indicated that *PFN2a* downregulated the protein content of CCNB1 (*p* < 0.01) and CCND1 (*p* < 0.05), and upregulated the protein level of p21, the cell proliferation negative regulator (*p* < 0.01) (Figure 2D,E). We concluded that *PFN2a* downregulates cell proliferation through increasing *p21* expression and decreasing *CCNB1* and *CCND1* expression. The results indicated suppression of *PFN2a* on C2C12 proliferation.

### 3.3. PFN2a Overexpression Inhibited C2C12 Myogenic Differentiation and Disturbed Sarcomere Structural Assembly

To further investigate the potential role of *PFN2a* in myogenic differentiation, WT and *PFN2a*-overexpressing cells were seeded in 6-well plates (8 × 10^4^/cm^2^) and cultured in the differentiation medium for seven days. The anti-MyHC immunofluorescent assay was used to measure the differentiation in WT and *PFN2a*-overexpressing cells on differentiation day 7 (Figure 3A). The immunofluorescent images and the quantitative data showed that *PFN2a* reduced (*p* < 0.001) the number of MyHC-positive cells (Figure 3A,B). In addition, *PFN2a* reduced the total number of cells in the case of the same number of cells on differentiation day 0 (Figure 3C). *PFN2a* significantly reduced the mRNA level of *MyHC* (*p* < 0.01) and *myogenin* (*p* < 0.01) (Figure 3D). Western blot results showed that *PFN2a* also downregulated the concentration of MyHC (*p* < 0.05) and myogenin (*p* < 0.01) (Figure 3E,F). The reduction of MyHC-positive cell number, and MyHC and myogenin content indicated that *PFN2a* downregulates C2C12 myogenic differentiation.

The mature sarcomere formation is a critical process for myogenic differentiation. Therefore, we explored the potential effect of *PFN2a* in sarcomere assembly in C2C12. WT and *PFN2a*-overexpressing cells were immunolabeled with antibodies against α-Actinin and MyHC on differentiation day 7. Immunohistochemistry data revealed that MyHC and α-Actinin were disordered in *PFN2a*-overexpressing cells (Figure 3G). The qPCR result showed that *PFN2a* significantly downregulated mRNA level of *α-actinin*, *tropomyosin 1* (*p* < 0.01), *titin* (*p* < 0.01), and *nebulin* (*p* < 0.01) on differentiation day 7 (Figure 3H). The results indicated that *PFN2a* disrupted the sarcomere assembly in the process of C2C12 myogenic development.

### 3.4. PFN2a Overexpression Promoted Apoptosis during C2C12 Myogenic Differentiation

To clarify whether the reduction in the total number of cells after overexpressed *PFN2a* (Figure 3C) in C2C12 myogenic differentiation is due to induction of apoptosis, the Annexin V-APC/7AAD Apoptosis Detection Kit was used to clarify if apoptosis was induced by *PFN2a* overexpression in the stage of differentiation (Figure 4A). *PFN2a* significantly decreased (*p* < 0.01, Figure 4B) the number of viable cells and increased early (*p* < 0.001, Figure 4C) and total apoptotic cells (*p* < 0.01, Figure 4D) on differentiation day 3. The qPCR results showed that *PFN2a* significantly upregulated mRNA levels of *caspase 8* (*p* < 0.05, Figure 4E) and *caspase 3* (*p* < 0.05, Figure 4E) and had no effect on *caspase 9* mRNA level (Figure 4E). *PFN2a* increased the protein level of caspase 3 (*p* < 0.05) and cleaved-caspase 3 (*p* < 0.01) (Figure 4F,G) on differentiation day 3. The results suggested that *PFN2a* promoted apoptosis in C2C12 myogenic differentiation.

### 3.5. PFN2a Overexpression Suppressed the Protein Content of HDAC1 in the Nucleus and Promoted the Content of p53 during C2C12 Myogenic Development

Previous studies have shown that PFN2a resided in the cytoplasm, interacted with HDAC1, and affected the nuclear localization of HDAC1. HDAC1 subsequently affected the activity and stability of p53. To explore whether *PFN2a* downregulation of C2C12 myogenic development is related to p53 and HDAC1, we performed qPCR and western blot to measure their mRNA levels and protein levels on the C2C12 proliferation day 1 (24h) and differentiation day 3. *PFN2a* significantly downregulated (*p* < 0.05, Figure 5A,B) mRNA levels of HDAC1 on the C2C12 proliferation day 1 and differentiation day 3. *PFN2a* significantly upregulated (*p* < 0.05, Figure 5A) mRNA levels of *p53* on the C2C12 proliferation day 1. *PFN2a* significantly upregulated (*p* < 0.01, Figure 5B) mRNA levels of *p53* on the differentiation day 3. Western blot results showed that *PFN2a* downregulated the protein level of HDAC1 (*p* < 0.01) and upregulated protein level of p53 (*p* < 0.05) on the C2C12 proliferation day 1 (Figure 5C,E). Western blot results showed that *PFN2a* decreased the protein level of HDAC1 (*p* < 0.05) and increased protein level of p53 (*p* < 0.05) on C2C12 differentiation day 3 (Figure 5D,F). These results showed that *PFN2a* decreased *HDAC1* expression, but increased *p53* expression during C2C12 myogenic development. To determine whether *PFN2a* regulates C2C12 myogenic development by affecting the subcellular localization of HDAC1, we performed western blot on proliferation day 1 and differentiation day 3. Western blot results showed PFN2a resided in the cytoplasm on proliferation day 1 and differentiation day 3 (Figure 5G,H). PFN2a did not affect HDAC1 level in the cytoplasm and decreased the amount of HDAC1 in the nucleus on proliferation day 1 and differentiation day 3 (Figure 5G,H). The results, therefore, imply that *PFN2a* suppresses the content of HDAC1 in the nucleus and then promotes the level of p53 during the C2C12 myogenic development.

### 3.6. PFN2a Overexpression Downregulated Myogenic Development through p53

Overexpressing *PFN2a* downregulated HDAC1 expression in the nucleus (Figure 5G,H) and increased *p53* expression (Figure 5C,D). Since p53 plays a critical role in cell proliferation and apoptosis and HDAC1 affected its ability in inducing growth arrest and apoptosis, we explored whether *PFN2a* regulates C2C12 myogenic development through p53. We successfully inhibited p53 expression in *PFN2a*-overexpressing cells by transfecting with si-*p53* (Appendix A). We performed EDU labeling and cell apoptosis assay to examine cell proliferation and apoptosis. We found that si-*p53* weakened (*p* < 0.05) the *PFN2a* regulatory effect on C2C12 proliferation (Figure 6A,B). si-*p53* significantly increased (*p* < 0.05, Figure 6D) the number of viable cells and decreased the number of early (*p* < 0.05, Figure 6E) and total apoptotic cells (*p* < 0.05, Figure 6F) in comparison with *PFN2a*-overexpressing cells on differentiation day 3. si-*p53* significantly weakened (*p* < 0.05) the *PFN2a* promotion effect on C2C12 apoptosis (Figure 6C) during differentiation. The MyHC immunofluorescent staining results showed that si-*p53* attenuated (*p* < 0.05, Figure 6H) *PFN2a* suppression effect on C2C12 differentiation day 7 (Figure 6G). In addition, si-*p53* increased the total cell number compared with *PFN2a*-overexpressing cells (Figure 6I). si-*p53* improved the myogenic development of *PFN2a*-overexpressing cells. Thus, we conclude that *PFN2a* overexpression slows down C2C12 myogenic development through p53 (Figure 7).

## 4. Discussion

Skeletal muscle maintains physical activity and energy homeostasis. A muscle development requires that myoblasts undergo myogenesis and immediately form new muscle fibers. Myogenesis of myoblasts includes proliferation, migration, differentiation, and fuse to form new muscle fibers [13]. PFN2a is a ubiquitous actin monomer-binding protein [29] and potentially participates in actin-based cellular biological processes, such as cell migration and proliferation [16]. Previous studies have found that HDAC1 knockdown induces cell apoptosis and inhibits cell proliferation [30,31]. In C2C12 cells, we found that *PFN2a* downregulates C2C12 proliferation (Figure 2) and promotes C2C12 apoptosis (Figure 4). Overexpression of *PFN2a* also downregulates the HDAC1 expression in the nucleus but increases p53 expression (Figure 5). PFN2a interacts with the full-length HDAC1 and C-HDAC1 fragments, but not the N-HDAC1 fragment, which blocks the NLS and suppresses the nuclear localization of HDAC1 [26]. Subsequent studies have found that HDAC1 deacetylates p53 and negatively regulates p53 stability and function in response to cellular stress [32]. However, p53 acetylation is indispensable for its stability and activation [33] and is correlated with its ability to induce growth arrest and apoptosis [27,32]. So, we explored whether *PFN2a* regulates C2C12 myogenic development through p53 by interfering *p53*. Si-*p53* reverses the *PFN2a* downregulation effect on C2C12 proliferation and the *PFN2a* promotion effect on C2C12 apoptosis at the stage of differentiation. Therefore, we concluded that *PFN2a* suppresses C2C12 myogenic development by decreasing C2C12 proliferation and promoting C2C12 apoptosis via inhibition of HDAC1 and the p53 pathway (Figure 7).

The proliferation of myoblasts governs skeletal muscles regeneration and the maintenance of muscle tissue. p53 is a short-lived protein [34] and remains at a low level as an inactive form [35]. In response to genotoxic stress, p53 activates and induces *p21* expression, which inhibits cyclin–CDK complexes and phosphorylation of Rb tumor suppressor gene, to ensure cell cycle arrest [36,37]. During the proliferation of C2C12 cells, cyclins and p21 control the cell cycle progression. *Cyclin B1* (*CCNB1*) is indispensable for cells to enter mitosis [38], while *cyclin D1* (*CCND1*) may have a leading role in G1/S phase regulation and influences G2/M transition [39]. Proliferating cell nuclear antigen (*PCNA*) is a standard marker of proliferation and participates in DNA replication [40]. p21 interacts with PCNA and prevents DNA replication [41,42]. p21 also interacts with cyclin–CDK complexes, thereby preventing CDKs activity and inhibiting cell cycle [43]. Previous studies have found that p53 regulates myoblast differentiation by means of pRb without affecting its cell cycle–related functions [44]. However, the cell cycle progression was controlled by multiple cyclin–Cdks. Cyclin D1 (CCND1) interacts with cyclin-dependent kinase-4/6 (Cdk4/6) to form the active Ccnd1-Cdk4/6 complex and regulates the G1-phase of the cell cycle [45]. Cyclin E (CCNE) interacts with Cdk2 to mediate the transition for the G1-S phase [46]. Cyclin A interacts with Cdk2 to regulate the progression of S-phase [47]. Cyclin B1 (CCNB1) or Cyclin A interacts with Cdk1 to mediate the transition for the G2-M phase [37]. So, p53-mediated cell cycle arrest regulates the activity of multiple cyclin-CDK complexes activities. We found that *PFN2a* downregulates cell proliferation through increasing *p21* expression and decreasing *CCNB1* and *CCND1* expression. *PFN2a* also downregulates the expression of HDAC1 in the cell nucleus and promotes *p53* expression. Si-*p53* reverses the *PFN2a* suppression effect on C2C12 proliferation. The results indicate that *PFN2a* suppresses C2C12 proliferation by directly reducing the content of HDAC1 in the nucleus, increasing *p53* expression, and ultimately inhibiting cell cycle progression through p21 (Figure 7).

Myoblast differentiation is critical to new myofiber formation. During C2C12 myogenic differentiation, there are three types of cells. Some C2C12 cells go to apoptosis; some undergo terminal differentiation to form myotubes; and some exit the cell cycle to enter a quiescent state for developing reserve cells [12]. The number of myoblasts participating in cell fusion is critical to terminal differentiation in myotube formation. Apoptosis mediates myogenesis by affecting the number of myoblasts [13]. p53 plays a crucial role in integrating intracellular signaling networks for mediating cellular responses to stress, such as oxidative stress and DNA damage [27]. In response to stress, the p53-mediated cell cycle arrests or induces cell apoptosis. As a “cellular gatekeeper” [48], p53 participates in two distinct apoptotic signaling pathways that are extrinsic and intrinsic apoptotic pathways and finally mediates apoptosis by activating the aspartate-specific cysteine proteases (Caspases) [34]. Caspase-3 plays a critical role in inducing apoptosis. Therefore, identifying the regulator in mediating apoptosis is vital for myogenic development. As for the proliferation of C2C12 decreased in *PFN2a* over-expressing cells (Figure 2), WT and *PFN2a*-overexpressing cells were plated with same high density on differentiation day 0 to avoid the effects of cell number on differentiation. When these cells were attached, we induced differentiation for the detection of late apoptosis and cell differentiation. We found that *PFN2a* promotes a cascade of expression of caspases, including *caspase 8* and *caspase 3*, during C2C12 myogenic differentiation. Apoptosis assay revealed that *PFN2a* overexpression promotes C2C12 cell apoptosis. *PFN2a* overexpression also decreases the HDAC1 content in the nucleus and increases *p53* expression. si-*p53* reverses the *PFN2a* promotion effect on C2C12 apoptosis on differentiation day 3. The results indicate that *PFN2a* promotes C2C12 apoptosis through inhibition of HDAC1 and the p53 pathway. Immunohistochemical analysis and qPCR revealed that *PFN2a* interferes the sarcomere assembly through indirect suppression of the mRNA levels of *α-actinin*, *titin*, *nebulin*, and *tropomyosin 1*, and consequentially inhibits C2C12 myogenic differentiation. Previous study p53-defective myoblasts do not differentiation [44]. However, *PFN2a* function of promoting C2C12 cell apoptosis via the p53 pathway leads to a decrease in the number of cells involved in differentiation (Figure 5B). So, si-*p53* improves the myogenic development of *PFN2a*-overespressing cells on differentiation day 7. These results indicate that inhibition of HDAC1 by PFN2a inhibits myogenic differentiation by promoting apoptosis via p53 in C2C12 (Figure 7).

In conclusion, our findings have further characterized the regulatory function of the gene *PFN2a*. *PFN2a* overexpression downregulates C2C12 myogenic development by inhibition of HDAC1 and proliferation inhibition and apoptosis promotion via the p53 pathway. Our results lead us propose a regulatory mechanism model how *PFN2a* regulates muscle development. PFN2a may be a therapeutic target for muscle disease treatment.

## Figures and Tables

**Figure 1 cells-08-00959-f001:**
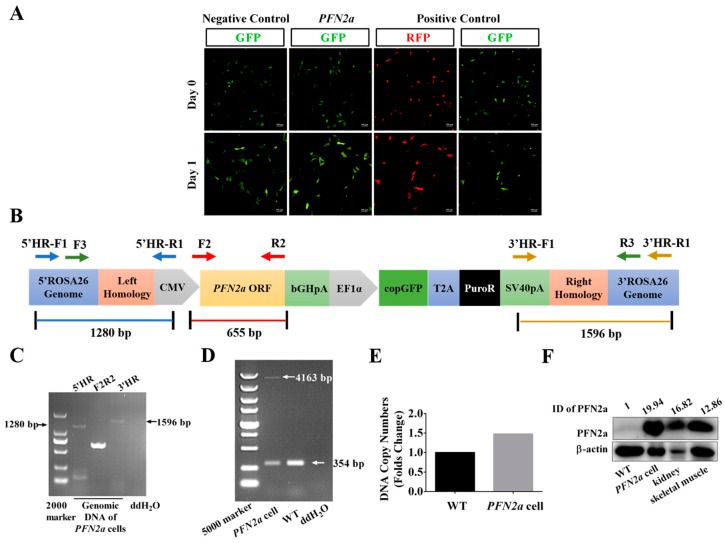
*PFN2a* knock-in at ROSA26 locus of C2C12 cells. (**A**) Monoclonal cell screening. The MCP-*ROSA26*-CG01 vector was transfected into C2C12 cells with DC-DON-SH02, DC-DON-SH02-*PFN2a* (*PFN2a* donor), and DC-RFP-SH02, respectively. The MCP-*ROSA26*-CG01 vector encoded Cas9 and gRNA. DC-DON-SH02 as a donor vector encoded green fluorescent protein (GFP). DC-DON-SH02 was used for constructing the *PFN2a* donor. The *PFN2a* donor contained the open reading frame (ORF) of *PFN2a*. *PFN2a* donor encoded GFP and PFN2a. The DC-RFP-SH02 vector encoded red fluorescent protein (RFP). The negative control group was transfected with DC-DON-SH02 and MCP-*ROSA26*-CG01 plasmids to establish transfection efficiency. The experimental group was transfected with *PFN2a* donor and MCP-*ROSA26*-CG01 plasmids to integrate *PFN2a* into the *ROSA26* locus. The positive control group was transfected with DC-RFP-SH02 and MCP-*ROSA26*-CG01 plasmids to establish fixed-point integration. (**B**) Primers used to identify the homologous recombination and genotype in *PFN2a*-overexpressing cells. Arrows show the direction and position of PCR primers. Primers used to identify the *PFN2a*-overexpressing cell line are shown in Table 2. (**C**) PCR results demonstrated the integration of the *PFN2a* donor at the *ROSA26* locus. Genomic DNA was extracted for PCR amplification from monoclonal cells. Primers of 5’HR and 3’HR were used to identify the integration of the *PFN2a* donor at the *ROSA26* locus. F2R2 primer was used to identify the integration of the open reading frame (ORF) of *PFN2a* at the *ROSA26* locus. Arrows show the size of the primers amplified fragment. (**D**) PCR results indicated monoclonal cells represented a heterozygous *PFN2a* cell line. Primers of F3R3 were used to amplify the 354-bp wild-type allele and the 4163-bp *PFN2a* donor allele in the genomic DNA from monoclonal cells. (**E**) DNA copy number of *PFN2a* in monoclonal cells was 0.5 times compared to WT. The absolute quantitative PCR analysis of *PFN2a* in monoclonal and WT cells. (**F**) Western blot identified the protein level of PFN2a in WT, *PFN2a* cell, kidney, and skeletal muscle from C57BL/6J mice. This monoclonal cell expressed PFN2a. β-actin was used as protein loading controls. Band intensities were quantified by Image J software and normalized to β-actin. Data were expressed as a fold change relative to WT. Magnification was 100×. The scale bar on the photomicrographs represented 100 μm. WT: wild type C2C12 cells; *PFN2a* cell: *PFN2a*-overexpressing C2C12 cells; HR: homologous recombination; ID: integrated density.

**Figure 2 cells-08-00959-f002:**
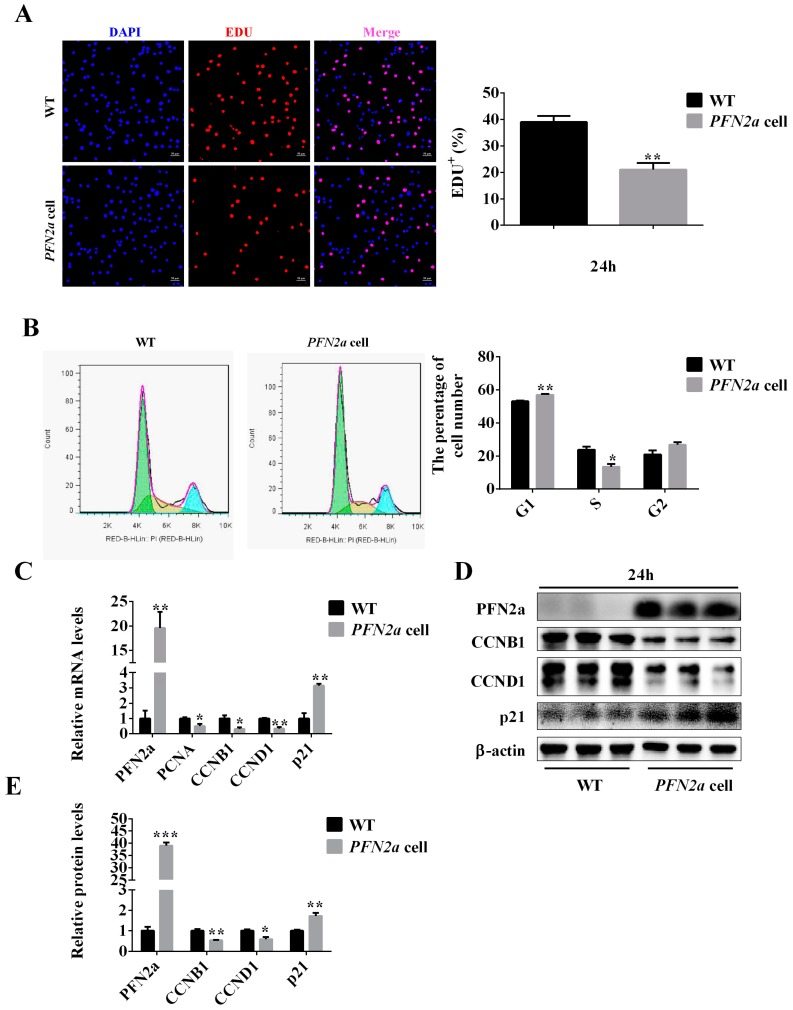
*PFN2a* overexpression inhibited C2C12 proliferation. (**A**) Representative images of the EDU staining for proliferating C2C12 are shown. The cell nucleus was stained with 4’,6-diamidino-2-phenylindole (DAPI). Proliferating C2C12 cells were labeled with 5-Ethynyl-2’-deoxyuridine (Edu) fluorescent dye (red). (**B**) The cell cycle in WT and *PFN2a*-overexpressing cells cultured for 24 h was analyzed by flow cytometry. The cells in the first green peak were in the G1 phase. The cells in the second yellow peak were in the S phase. The cells in the third blue peak were in the G2/M phase. (**C**) qPCR results indicated that *PFN2a* downregulated the mRNA level of *CCNB1*, *CCND1*, and *PCNA*, and increased the *p21* mRNA level. (**D**) Western blot results showed the protein levels of cell cycle-related proteins and p21 corresponded to the qPCR results. (**E**) The relative protein levels obtained through WB band gray scanning showed that *PFN2a* downregulated the protein level of CCNB1 and CCND1 and upregulated the protein level of p21. *, statically significant at *p* < 0.05; **, statically significant at *p* < 0.01; ***, statically significant at *p* < 0.001. The results were presented as mean ± S.E.M. of three replicates for each group. The statistical significance of differences between means was assessed using unpaired Student’s *t*-test. Magnification was 200×. The scale bar on the photomicrographs represents 50 μm. WT: wild type C2C12 cells; *PFN2a* cells: *PFN2a*-overexpressing C2C12 cells.

**Figure 3 cells-08-00959-f003:**
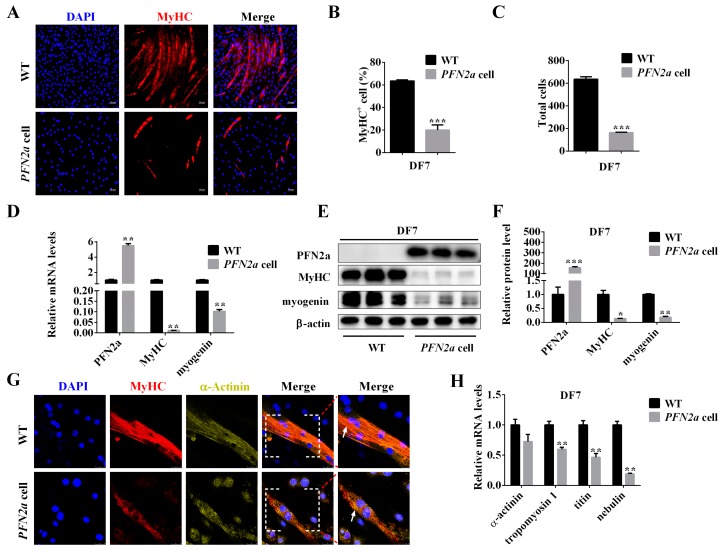
*PFN2a* overexpression inhibited C2C12 myogenic differentiation and disrupted the sarcomere structure assembly. (**A**) Representative images of the MyHC immunofluorescent staining for differentiated C2C12 and *PFN2a*-overexpressing cells on differentiation day 7. MyHC: red, a molecular marker of myogenesis; DAPI: blue, cell nucleus; Merge: the cell fused into myotubes. (**B**) *PFN2a* decreased the percentage of MyHC-positive cells. (**C**) The total number of cells in magnification 200× on differentiation day 7. (**D**) The qPCR results confirmed that *PFN2a* downregulated mRNA level of *MyHC* and *myogenin* on differentiation day 7. (**E**) Western blot analysis of MyHC and myogenin after differentiation induction. (**F**) The relative protein levels obtained through WB band gray scanning showed that *PFN2a* downregulated the content of MyHC and myogenin. (**G**). Representative images of the MyHC and α-Actinin immunofluorescent staining for visualizing sarcomere on differentiation day 7. MyHC assembled to form thick filament (Red). α-Actinin located at Z disk (Yellow). *PFN2a* disrupted sarcomerogenesis. (**H**) The qPCR results indicated that *PFN2a* reduced mRNA level of sarcomerogenesis marker genes, *α-actinin*, *tropomyosin 1*, *titin*, and *nebulin* on differentiation day 7. The representative images of the MyHC immunofluorescent staining magnified 200×. The scale bar on these photomicrographs represented 50 μm. The representative images of the sarcomere magnified 630×. The scale bar on these photomicrographs represented 25 μm. The scale bar on these photomicrographs from the virtual frame area represents 10 μm. The arrows indicated sarcomere. Overlapping areas between α-Actinin (Yellow) and MyHC (Red) appeared orange. The results were presented as mean ± S.E.M. of triplicate experiments for each group. The statistical significance of differences between means was assessed using unpaired Student’s *t*-test. *, *p* < 0.05; **, *p* < 0.01, ***, *p* < 0.001. WT: wild type C2C12 cells; *PFN2a* cell: *PFN2a*-overexpressing C2C12 cells; DF: differentiation.

**Figure 4 cells-08-00959-f004:**
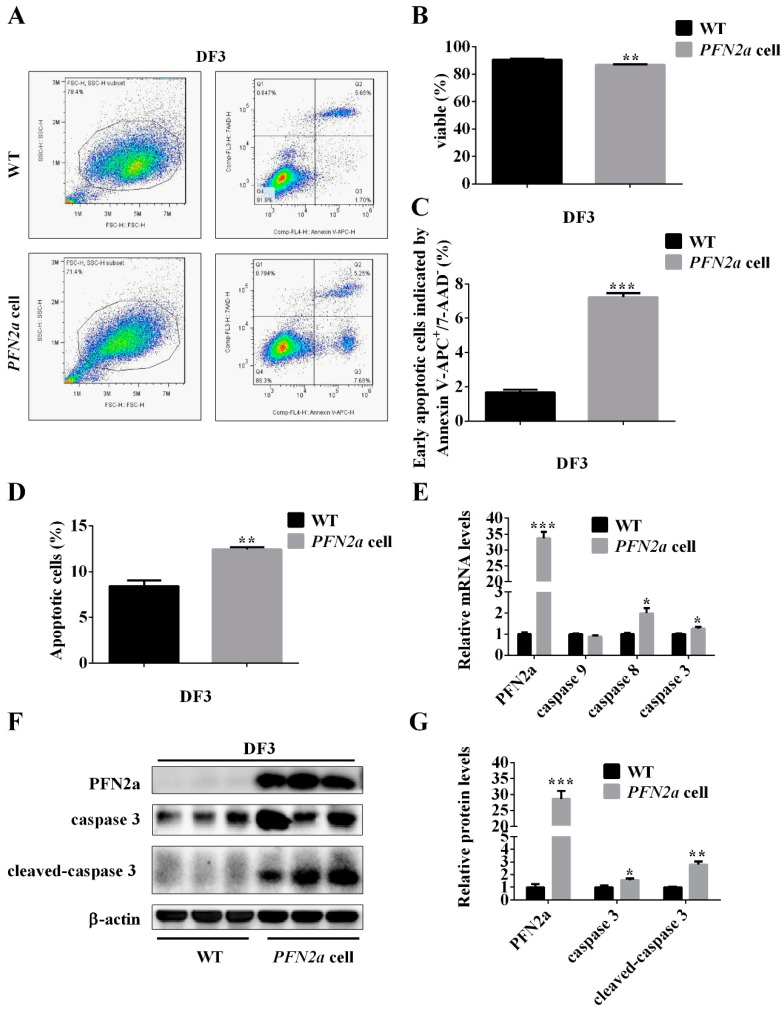
*PFN2a* overexpression promoted apoptosis during C2C12 myogenic differentiation. (**A**) Representative images of apoptosis assay for C2C12 cells and *PFN2a* cells on differentiation day 3 by using the Annexin V-APC/7AAD Apoptosis Detection Kit. The cells gated in Q2 quadrant displayed Annexin V-APC-positive/7AAD-positive, late apoptotic cells. Cells localized in Q3 quadrant displayed Annexin-V-APC-positive/7AAD-negative, early apoptotic cells. Cells localized in Q4 displayed Annexin-V-APC-negative/7AAD-negative, viable cells. (**B**) *PFN2a* suppressed the number of viable cells on differentiation day 3. (**C**) *PFN2a* increased early apoptotic cells on differentiation day 3. (**D**) *PFN2a* increased total apoptosis cells on differentiation day 3. (**E**) The qPCR results confirm *PFN2a* increased mRNA level of apoptosis-related genes, *caspase 8* and *caspase 3,* on differentiation day 3. (**F**) Western blot analysis of apoptosis-related proteins, caspase 3, and cleaved-caspase 3 on differentiation day 3. (**G**) The relative protein levels obtained through WB band gray scanning analysis on differentiation day 3. The results were presented as mean ± S.E.M. of triplicate experiments for each group, and statistical significance of differences between means was assessed using unpaired Student’s *t*-test (*, *p* < 0.05; **, *p* < 0.01; ***, *p* < 0.001). WT: wild type C2C12 cells; *PFN2a* cell: *PFN2a*-overexpressing C2C12 cells; DF: differentiation.

**Figure 5 cells-08-00959-f005:**
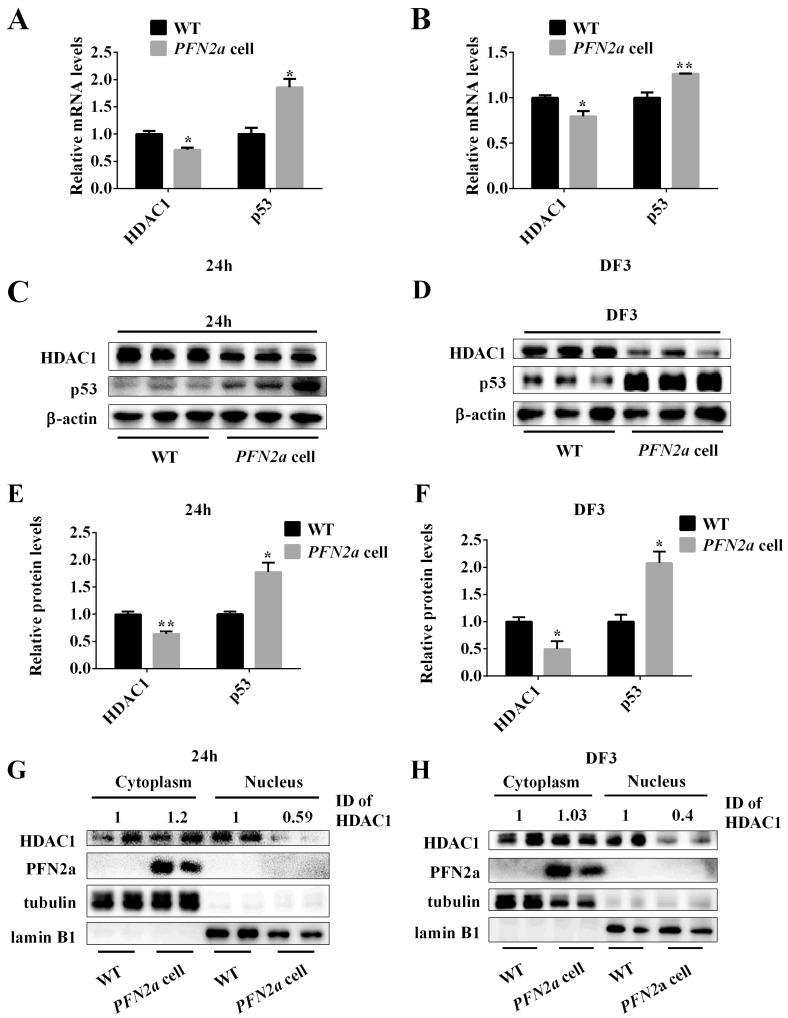
*PFN2a* overexpression downregulated the content of HDAC1 in the nucleus, but upregulated the level of p53 during the C2C12 myogenic development. (**A**,**B**) *PFN2a* downregulated the mRNA level of *HDAC1* and upregulated mRNA level of *p53* on proliferation day 1 and differentiation day 3. (**C**,**D**) Western blot results showed that the protein level corresponded to the mRNA result. (**E**,**F**) The relative protein level of HDAC1 and p53 obtained through WB band gray scanning analysis on proliferation day 1 and differentiation day 3. (**G**,**H**) *PFN2a* reduced the nuclear content of HDAC1 on proliferation day 1 and differentiation day 3. Lamin B1 and tubulin were used as nuclear- or cytoplasmic-specific protein loading controls, respectively. The results are presented as mean ± S.E.M. of triplicate experiments for each group, and statistical significance of differences between means was assessed using unpaired Student’s *t*-test (*, *p* < 0.05; **, *p* < 0.01). Band intensities were quantified by Image J software and normalized to tubulin or lamin B1. Data were expressed as change in fold relative to the control. WT: wild type C2C12 cells; *PFN2a* cell: *PFN2a*-overexpressing C2C12 cells. 24h: on proliferation day 1. DF: differentiation. ID: integrated density.

**Figure 6 cells-08-00959-f006:**
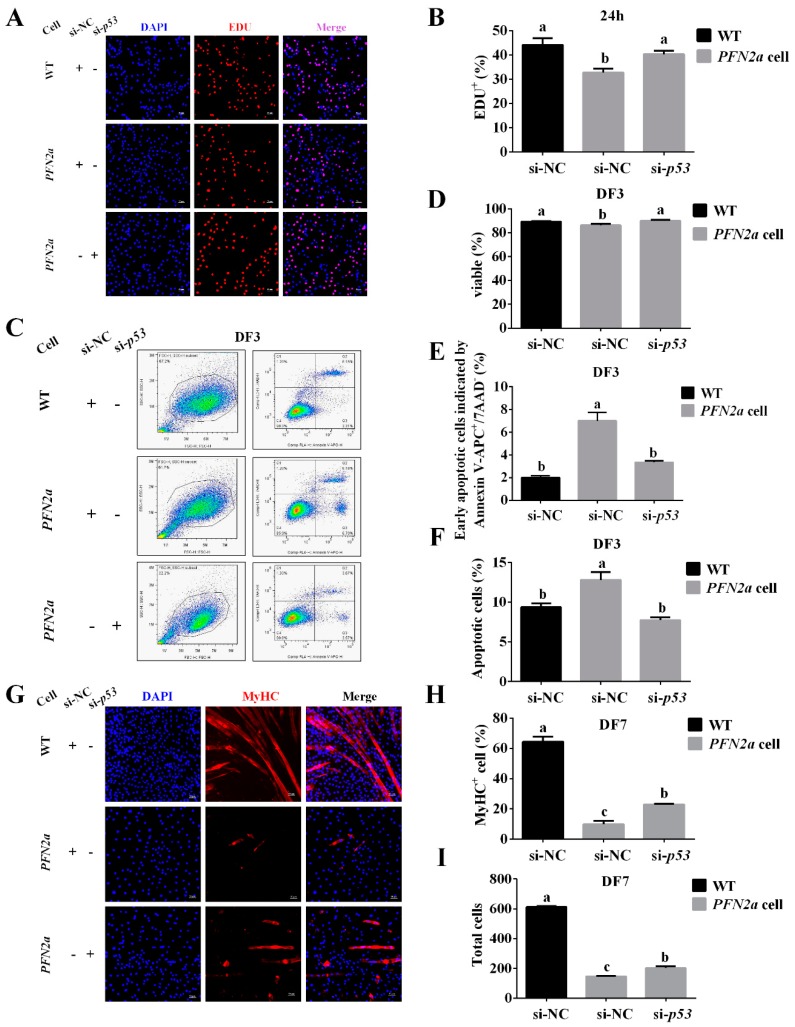
*PFN2a* overexpression suppressed myogenic development through p53. (**A**) Representative images of the EDU staining for proliferating WT and *PFN2a* cells are shown. (**B**) EDU labeling analysis of the function of si-*p53* in abolishing the *PFN2a* effect on C2C12 proliferation. (**C**) Cell apoptosis assay analysis of the function of si-*p53* in attenuating the *PFN2a* promotion effect on C2C12 apoptosis at the stage of differentiation. (**D**) si-*p53* increased the number of viable cells in comparison with *PFN2a* cells. (**E**) si-*p53* decreased early apoptotic cells compared with *PFN2a* cells. (**F**) si-*p53* decreased the total number of apoptotic cells compared with *PFN2a*-overexpressing cells on differentiation day 3. (**G**) Representative images of the MyHC immunofluorescent staining for differentiated WT and *PFN2a* cells on differentiation day 7. (**H**) Anti-MyHC immunofluorescent assay showed the si-*p53* effect on *PFN2a* expression to inhibit C2C12 differentiation on differentiation day 7. *PFN2a* decreased the percentage of MyHC-positive cells. (**I**) The total number of cells in magnification 200× on differentiation day 7. The results were presented as mean ± S.E.M. of triplicate experiments for each group. The data are in a normal distribution; and homogeneity of data between each treatment group is equal by SPSS analysis. Significant differences between treatment groups were determined by one-way ANOVA (SPSS v18.0, IBM Knowledge Center, Chicago, IL, USA). *p* < 0.05 was considered statistically significant. WT: wild type C2C12 cells; *PFN2a* cell: *PFN2a*-overexpressing C2C12 cells; 24h: on proliferation day 1; DF: differentiation; Si-NC: siRNA-negative control; Si-*p53*: siRNA-*p53*.

**Figure 7 cells-08-00959-f007:**
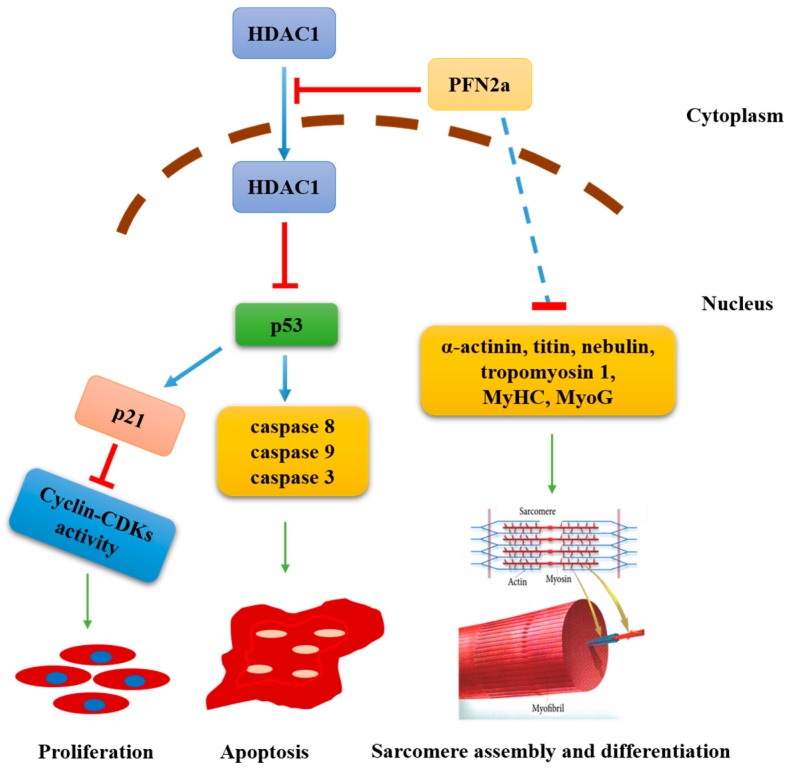
*PFN2a* overexpression downregulates myogenic development via the p53 pathway in C2C12 cells. We constructed a C2C12 *PFN2a*-overexpressing cell line through the CRISPR/Cas9 system. During C2C12 myogenic development, *PFN2a* decreased the content of HDAC1 in the nucleus and upregulated *p53* expression. *PFN2a* downregulated C2C12 proliferation and promoted apoptosis through p53 by directly reducing the content of HDAC1 in the nucleus. *PFN2a* destructed the sarcomere structure assembly by indirect downregulation on the mRNA levels of *α-actinin*, *titin*, *nebulin*, and *tropomyosin 1*. Inhibition of proliferation and promotion of apoptosis led to differentiation inhibition and sarcomere structure assembly destruction.

**Table 1 cells-08-00959-t001:** Primers used for vector construction. ORF: open reading frame.

Primer Name	Primer Sequences ^1^ (5′ to 3′)	Size (bp)	Application
*PFN2a* donor primer	F: A**GGCGCGCC**ACCGCCTCTGCTCCTGC	673	Amplification of the ORF of *PFN2a* for constructing *PFN2a* donor
	R: CGC**GGATCC**CCGCCTCTAACCAATGCTG		
pCDNA3.1 (+)-*PFN2a primer*	F: GG**GGTACC**ATGGCCGGTTGGCAGAGCTA	439	Amplification of the ORF of *PFN2a* for constructing pCDNA3.1 (+)-*PFN2a*. This vector was used to establish standard curve line for absolute quantitative analysis of *PFN2a*- overexpressing cell line.
	R: CG**GAATTC**CTAGAACCCAGAGTCTCTCAAGT		

^1^ Sequences in bold represent the enzyme cutting sites.

**Table 2 cells-08-00959-t002:** Primers used for identifying the *PFN2a*-overexpressing cell line.

Primer Name	Primer Sequences ^1^ (5′ to 3′)	Size (bp)	Application
5′HR	F: AGGGAGCGGAAAAGTCTCCA	1280	Accurate detection of integration of *PFN2a* donor into *ROSA26* locus.
	R: CTCCCACCGTACACGCCTAC		
3′HR	F: GGTGCCTGAAATCAACCTCTGGA	1596	
	R: TCAAGCCAGTCCAAGAGAAAGCA		
F2R2	F2: ACCGCCTCTGCTCCTGC	655	Amplification of the ORF of *PFN2a*.
	R2: CCGCCTCTAACCAATGCTG		
F3R3	F3: CAGGCAGACCTCCATCGC	354 or 4163	Genotype analysis of *PFN2a*-overexpressing cell line by PCR.
	R3: GACAACGCCCACACACCAG		
F4R4	F: CAACGGCAAAATACTTGAGAGA	120	Genotype analysis of *PFN2a*-overexpressing cell line by absolute quantitative analysis
	R: CCGACGGATACAAAGGAGAC		

**Table 3 cells-08-00959-t003:** Primers used for qPCR.

Primer Name	Primer Sequences ^1^ (5′ to 3′)	Size (bp)	Accession Number
*PFN2a*	F: CAACGGCAAAATACTTGAGAGA	120	NM_019410.3
	R: CCGACGGATACAAAGGAGAC		
*PCNA*	F: AGCGGAGAAGGTGCTGGAG	157	NM_011045.2
	R: ATAGCGGCGGTATGTGTCG		
*CCNB1*	F: TAGGGCGAGGTCAGTATGGC	211	NM_172301.3
	R: ACTTCCCGACAGGTTTTGGTAG		
*CCND1*	F: GCCACGCCCTCCGTATCT	190	NM_007631.2
	R: GTAACCAGCGGCTCTTCTTCA		
*p21*	F: CGAGAACGGTGGAACTTTGAC	107	NM_007669.5
	R: CCAGGGCTCAGGTAGACCTT		
*MyHC*	F: CCCGCTAAGGGTCTTCGTA	176	NM_030679.2
	R: GCCCCGTTGACATTGGA		
*myogenin*	F: CCTGGAAGAAAAGGGACTGG	246	NM_031189.2
	R: CGCTCAATGTACTGGATGGC		
*α-actinin*	F: ATGCCTCGCTGCTGAATGA	191	NM_134156.2
	R: GCTCGCAAAAGCCTCGTG		
*tropomyosin 1*	F: AACCCGCACAAATACTCCGA	138	NM_001164248.1
	R: AACAGCCCACTCCTCCTCAAC		
*titin*	F: GCCGCCTGGAATCCCTAC	324	NM_011652.3
	R: ACGCATCTGGCATCAAAGTG		
*nebulin*	F: CAGCCAAAGCCACCCCA	348	NM_010889.1
	R: TCTCACCAACCCGCCTCAT		
*caspase 9*	F: CGCCATCTGGGTCTCGG	241	NM_001277932.1
	R: ACTCGCTGCTCCTTTGCTG		
*caspase 8*	F: CGCCCGTGCTTGGACTAC	152	NM_001080126.1
	R: TTCTCCCGCCGACTGATGT		
*caspase 3*	F: GTCTGACTGGAAAGCCGAAACT	104	NM_001284409.1
	R: GCAAGCCATCTCCTCATCAGTC		
*p53*	F: GAACCGCCGACCTATCCTTA	95	NM_001127233.1
	R: GGCAGGCACAAACACGAAC		
*HDAC1*	F: AGTCTGTTACTACTACGACGGG	101	NM_008228.2
	R: TGAGCAGCAAATTGTGAGTCAT		
*β-actin*	F: TGGTGCGAATGGGTCAGAA	310	NM_007393.5
	R: CCGCCAGAGGCATACAGG		

**Table 4 cells-08-00959-t004:** Primary antibodies used in the study.

Primary Antibody	Clone	Company	Catalog No.	Dilution
PFN2a	Polyclonal	Enogene	E93047	1:1000
CCNB1	Polyclonal	CST	4138	1:1000
CCND1	Polyclonal	CST	2922	1:1000
MyHC	Monoclonal	BOSTER	BM0096	1:500
myogenin	Polyclonal	SANTA CRUZ	J2314	1:200
α-Actinin	Polyclonal	GeneTex	GTX103219	1:1000
caspase 3 (H-277)	Polyclonal	SANTA CRUZ	sc-7148	1:1000
cleaved caspase 3 (Asp175)	Polyclonal	CST	9661	1:1000
p53	Monoclonal	GeneTex	GTX70214	1:5000
HDAC1	Polyclonal	Bioss	bs-1414R	1:1000
tubulin	Polyclonal	Bioworld	AP0064	1:5000
lamin B1	Polyclonal	Bioworld	AP6001	1:5000
β-actin	Monoclonal	Bioworld	BS6007M	1:5000

**Table 5 cells-08-00959-t005:** Secondary antibody used in the study.

Secondary Antibody	Conjugate Used	Company	Catalog No.	Dilution
Goat anti-Rabbit IgG ^1^ (H+L)	HRP ^2^	Bioworld	BS13278	1:50000
Goat anti-Mouse IgG ^1^ (H+L)	HRP ^2^	Bioworld	BS12478	1:50000
Goat Anti-Mouse IgG ^1^ (H+L)	Cy3 ^3^	Bioworld	BS10006	1:200
Goat Anti-rabbit IgG ^1^	Alexa Fluor 647 ^3^	Bioss	bs-0295G-AF647	1:200

^1^ Goat polyclonal secondary antibody to rabbit/mouse immunoglobulin G (IgG). ^2^ Horseradish peroxidase (HRP) conjugated anti-rabbit/mouse secondary antibody optimized for western blot. ^3^ Cy3 and Alexa Fluor 647 are fluorescent dyes that are particularly suitable for protein labeling. Cy3 and Alexa Fluor 647 are conjugated anti-rabbit/mouse secondary antibody optimized for immunofluorescent staining.

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
