# Peer review of "PFN2a Suppresses C2C12 Myogenic Development by Inhibiting Proliferation and Promoting Apoptosis via the p53 Pathway"

_cells, 2019, doi:10.3390/cells8090959_

Round 1

Reviewer 1 Report

The manuscript by Li H et al., describes that Up-expression of PFN2a suppresses C2C12 myogenic development through p53 pathway proliferation inhibition and apoptosis promotion. Basically, the role of PFN2a in myogenic development presented in this manuscript are interesting and well organized. This has an important information for the molecular cell science. For the benefit of the reader, however, some points need clarifying. My comments to this article were as follows.

The authors showed that PFN2a suppresses C2C12 myogenic development through p53 pathway in this experiment. Comparison to previous report that p53 regulates myoblast differentiation by means of pRb without affecting its cell cycle–related functions. (Porrello A et al., J Cell Biol. 2000 Dec 11;151(6):1295-304. p53 regulates myogenesis by triggering the differentiation activity of pRb.), there are some discrepancy. The authors should describe in discussion. The authors described in introduction that the profilin expression is decreased during the progression of C2C12 myogenic differentiation. Does expression of PFN2a decrease during the progression of C2C12 myogenic differentiation? In line 140, it does not seem that conclusion of suppression of PFN2a on C2C12 prolifereation in the result of Fig.2. In Fig 3, the authors were used PNF2a-overexpressing myoblasts. When PNF2a is transfected in differentiated myotubes, how is the result of differentiation.

Author Response

1. The authors showed that PFN2a suppresses C2C12 myogenic development through p53 pathway in this experiment. Comparison to previous report that p53 regulates myoblast differentiation by means of pRb without affecting its cell cycle–related functions. (Porrello A et al., J Cell Biol. 2000 Dec 11;151(6):1295-304. p53 regulates myogenesis by triggering the differentiation activity of pRb.), there are some discrepancy. The authors should describe in discussion.

Thanks for your comment. We have added the related description of p53-mediated cell cycle arrest (lines 323-324, 330-339) and p53-mediated differentiation (367-369) in Discussion. We have added four related references (#36, #43, #44, #45, and #46) in the discussion section (lines 324, 332-337, and 368). These new changes are marked in red.

2. The authors described in introduction that the profilin expression is decreased during the progression of C2C12 myogenic differentiation. Does expression of PFN2a decrease during the progression of C2C12 myogenic differentiation?

Thank you very much for this valuable suggestion. We apologize for the unclear description that may not be clear to all the readers. We examined the PFN2a expression during the progression of C2C12 myogenic differentiation using qPCR and western blot (Fig. S2). We have added the related details in the Results (lines 87-88) and the Supplementary Materials (lines 527-540, Figure S2). These new changes are marked in red.

3. In line 140, it does not seem that conclusion of suppression of PFN2a on C2C12 proliferation in the result of Fig.2.

Thanks for pointing out the sentence that may not be clear to all the readers. We have added the related details in Results (lines 134-146). The new changes are marked in red.

4. In Fig 3, the authors were used PFN2a-overexpressing myoblasts. When PFN2a is transfected in differentiated myotubes, how is the result of differentiation.

We inserted PFN2a into C2C12 cells at ROSA26 locus using CRISPR/Cas9 for constructing stable PFN2a-overexpressing C2C12 cells (PFN2a cell). PFN2a cells are induced to differentiate. Results show PFN2a inhibited C2C12 myogenic differentiation (See Figure 3).

Reviewer 2 Report

The study submitted by Huaqin Li, seek to elucidate the role of PFN2a in the myoblast myogenic development. The authors demonstrate that PFN2a overexpression downregulates C2C12 myogenic development by inhibition of HDAC1 and proliferation inhibition and apoptosis promotion through p53 cascade. The manuscript is well written, the methods appear appropriate and utilized in an appropriate manner to examine questions posed. The methods are described with sufficient detail that they could be repeated.  The discussion is reasonable and focused on the interpretation of the results in context of PFN2a overexpression affects myogenic development in C2C12 cells, involving p53 ad HDAC1. However, there are certain points that could be improved:

1- The title, although very descriptive, is long. The authors could present the same quality of information in the title but in a more concise way.

2- Figure 3A. The merge does not seem to be the combination of the two figures shown. It is probably a problem of image quality / size. The same with figure 2, images of the EDU staining are of poor quality.

3- The authors base their results on experiments performed with the C2C12 cell line. The work would be more relevant if some of the tests are carried out in primary myoblast cultures and evaluate whether the result is comparable to that obtained with c2c12.

4- Figure 5, the images of the blots (G and H) are pixelated – Please improve its quality.

5- In Methods, please mention the passage numbers of the cells used in each assay. Cell line responses are dependent of the passage numbers.

Author Response

1. The title, although very descriptive, is long. The authors could present the same quality of information in the title but in a more concise way.

We have modified the title to “PFN2a suppresses C2C12 myogenic development by inhibiting proliferation and promoting apoptosis via p53 pathway” per your suggestion (see lines 2-4).

2. Figure 3A. The merge does not seem to be the combination of the two figures shown. It is probably a problem of image quality / size. The same with figure 2, images of the EDU staining are of poor quality.

Thanks for pointing out the problem of poor image quality that may not be clear to all the readers. We have modified the Figure 3A (see line 182) and Figure 2 (see line 148). We will upload the high definition version of the data map.

3. The authors base their results on experiments performed with the C2C12 cell line. The work would be more relevant if some of the tests are carried out in primary myoblast cultures and evaluate whether the result is comparable to that obtained with c2c12.

Thank you very much for this valuable suggestion. To our knowledge, there is no paper reported the regulatory function of PFN2a during C2C12 myogenic differentiation. So, we studied the function of PFN2a in C2C12 cells first. We have examined the expression levels of PFN2a in skeletal muscle of mice at different embryonic age and different developmental stages after birth. These results will be published along with the results of our ongoing transgenic mice and future primary cell validation.

4. Figure 5, the images of the blots (G and H) are pixelated – Please improve its quality.

Thanks for pointing out the problem of poor image quality that may not be clear to all the readers. The image pixels are lower in the Word file. We have modified the Figure 5G and 5H (see line 255). We will upload the high definition version of the data map.

5. In Methods, please mention the passage numbers of the cells used in each assay. Cell line responses are dependent of the passage numbers.

Thank you for your suggestion. We have added this information in the Method section (see lines 392, 405, 418-419, 442, 450, 458, 479-480, and 486-487). These new changes are marked in red.

Reviewer 3 Report

The study by Li and colleagues establishes the role of PFN2a in myogenic differentiation. In particular, using an in-vitro model of PFN2a over-expression, they evaluate multiple aspects such as cell proliferation, induction of apoptosis and the differentiation itself, providing new insight into the biological functions of PFN2a.

From my point of view, I think that the publication in this journal is appropriate. Nevertheless, I would suggest some experiments to the authors in order to improve the manuscript:

-Since proliferation of C2C12 was strongly decreased in PFN2a over-expressing cells as compared with that of WT, the following differentiation/fusion phase was consequently altered (less cells, less myotubes). Specific differentiation assay, during which the cells are not firstly expanded, should be performed (plated directly with high density).

-The authors state that the effect of the PFN2a over-expression is mediated by the inhibition of HDAC1 activity. To better untangle these events, I suggest to pharmacologically or genetically inhibit HDAC1 to obtain similar results.

-Since all the experiments shown by the authors are carried out in a condition of PFN2a overexpression, to better understand the physiological role of this mechanism, it would be useful to show the transcriptional changes and the protein levels of the players (PFN2a, HDAC1, p53) during the proliferation phases and at different times of cell differentiation. It would also be interesting to check the activity of these factors in conditions of stress (for example after treatment with hydrogen peroxide).

Author Response

1. Since proliferation of C2C12 was strongly decreased in PFN2a over-expressing cells as compared with that of WT, the following differentiation/fusion phase was consequently altered (less cells, less myotubes). Specific differentiation assay, during which the cells are not firstly expanded, should be performed (plated directly with high density).

Thanks for pointing out the experimental design that may not be clear to all the readers. We have added this information in the Discussion (see lines 355-359) and Method (see lines 396-398 and 487-488). These new changes are marked in red.

2.The authors state that the effect of the PFN2a over-expression is mediated by the inhibition of HDAC1 activity. To better untangle these events, I suggest to pharmacologically or genetically inhibit HDAC1 to obtain similar results.

Thank you very much for your suggestion. Previous studies have found that PFN2a suppressed the nuclear localization of HDAC1 [1]. And HDAC1 affected the activity of p53 by changing p53 acetylation state and finally induced p53 degradation, with alterations of p53 target gene [2] and participated in cell growth, and apoptosis (see lines 71-76). We found overexpressing PFN2a also downregulated HDAC1 expression in the nucleus (Figure 5G, 5H) and increased p53 expression (Figure 5C, 5D). So, we explored whether PFN2a regulates C2C12 myogenic development through p53 by interfering p53. We have modified the description as to this comment in the Results (lines 270-272) and the Discussion (see lines 308-310 and 315-316). The new changes are marked in red on lines 306-308 and 315-316.

3. Since all the experiments shown by the authors are carried out in a condition of PFN2a overexpression, to better understand the physiological role of this mechanism, it would be useful to show the transcriptional changes and the protein levels of the players (PFN2a, HDAC1, p53) during the proliferation phases and at different times of cell differentiation. It would also be interesting to check the activity of these factors in conditions of stress (for example after treatment with hydrogen peroxide).

Thank you very much for your suggestion. Previous studies have found HDAC1 affected p53 functions in cell growth and apoptosis (see lines 71-75). This study focused on the mechanisms which PFN2a regulates C2C12 myogenic development. We constructed stable PFN2a-overexpressing C2C12 cells (PFN2a cell) using the CRISPR/Cas9 system. We used this stable PFN2a-overexpressing C2C12 cells to study the function of PFN2a in C2C12 myogenic development. We have clarified that PFN2a suppresses C2C12 myogenic development by inhibiting proliferation and promoting apoptosis via the p53 pathway. As to the transcriptional change and the protein level of PFN2a, we have added the related details in the Results (lines 87-88) and the Supplementary Materials (lines 527-540, Figure S2). As to treatment with hydrogen peroxide (H2O2) in C2C12, this is a new aspect of research, and we will explore it in the next experiment.

References

Tang, Y.N.; Ding, W.Q.; Guo, X.J.; Yuan, X.W.; Wang, D.M.; Song, J.G. Epigenetic regulation of Smad2 and Smad3 by profilin-2 promotes lung cancer growth and metastasis. Nat Commun 2015, 6, 8230, doi:10.1038/ncomms9230. Mizuguchi, Y.; Specht, S.; Lunz, J.G., 3rd; Isse, K.; Corbitt, N.; Takizawa, T.; Demetris, A.J. SPRR2A enhances p53 deacetylation through HDAC1 and down regulates p21 promoter activity. BMC Mol Biol 2012, 13, 20, doi:10.1186/1471-2199-13-20.